# Combining rVAR2 and Anti-EpCAM to Increase the Capture Efficiency of Non-Small-Cell Lung Cancer Cell Lines in the Flow Enrichment Target Capture Halbach (FETCH) Magnetic Separation System

**DOI:** 10.3390/ijms25189816

**Published:** 2024-09-11

**Authors:** Sitian He, Peng Liu, Yongjun Wu, Mette Ø. Agerbæk, Ali Salanti, Leon W. M. M. Terstappen, Pascal Jonkheijm, Michiel Stevens

**Affiliations:** 1Department of Medical Cell Biophysics, TechMed Center, Faculty of Science and Technology, University of Twente, P.O. Box 217, 7500 AE Enschede, The Netherlands; hesitian135@126.com (S.H.); p.liu-2@utwente.nl (P.L.); or leonardus.terstappen@med.uni-duesseldorf.de (L.W.M.M.T.); 2College of Public Health, Zhengzhou University, Zhengzhou 450001, China; wuyongjun135@126.com; 3Department of Personalized Diagnosis & Therapeutics, Faculty of Science and Technology, University of Twente, P.O. Box 217, 7500 AE Enschede, The Netherlands; 4Department of Molecules and Materials, Laboratory of Biointerface Chemistry and the TechMed Centre, University of Twente, 7500 AE Enschede, The Netherlands; 5Centre for Translational Medicine and Parasitology at Department of Immunology and Microbiology, Faculty of Health and Medical Sciences, University of Copenhagen and Department of Infectious Diseases, Copenhagen University Hospital, 2200 Copenhagen, Denmark; mettea@sund.ku.dk (M.Ø.A.); salanti@sund.ku.dk (A.S.); 6VarCT Diagnostics, 2000 Frederiksberg, Denmark; 7Department of General, Visceral and Pediatric Surgery, Heinrich-Heine University, University Hospital Düsseldorf, 40225 Düsseldorf, Germany; 8FETCH BV, 7437AE Deventer, The Netherlands

**Keywords:** circulating tumor cells (CTCs), non-small-cell lung cancer (NSCLC), immunomagnetic enrichment, rVAR2, EpCAM

## Abstract

Circulating tumor cells (CTCs) are detected in approximately 30% of metastatic non-small-cell lung cancer (NSCLC) cases using the CellSearch system, which relies on EpCAM immunomagnetic enrichment and Cytokeratin detection. This study evaluated the effectiveness of immunomagnetic enrichment targeting oncofetal chondroitin sulfate (ofCS) using recombinant VAR2CSA proteins (rVAR2) to improve the recovery of different NSCLC cell lines spiked into lysed blood samples. Four NSCLC cell lines—NCI-H1563, A549, NCI-H1792, and NCI-H661—were used to assess capture efficiency. The results demonstrated that the combined use of anti-EpCAM antibody and rVAR2 significantly enhanced the capture efficiency to an average of 88.2% compared with 40.6% when using only anti-EpCAM and 56.6% when using only rVAR2. These findings suggest that a dual-marker approach using anti-EpCAM and rVAR2 can provide a more robust and sensitive method for CTC enrichment in NSCLC, potentially leading to better diagnostic and prognostic outcomes.

## 1. Introduction

Circulating tumor cells (CTCs) are tumor cells that detach from the primary tumor or metastatic lesions and enter the bloodstream [1,2]. The detection and isolation of CTCs hold great promise for predicting disease progression, anticipating treatment responses, and determining optimal treatment strategies through their in-depth characterization [3]. Currently, the CellSearch^®^ system is the only platform approved by the U.S. Food and Drug Administration (FDA) for the detection of CTCs [4]. This platform utilizes magnetic nanoparticles coated with anti-EpCAM antibodies to target EpCAM for identification of CTCs and subsequently conduct bulk magnetic enrichment. In approximately 30% of metastatic non-small-cell lung cancer (NSCLC) cases, CTCs can be detected using this EpCAM-targeting platform [5]. EpCAM is a cell surface glycoprotein typically overexpressed in epithelial tumor cells, but not expressed in tumors of mesodermal and ectodermal origin, such as neurogenic tumors, sarcomas, melanomas, or lymphomas, and is downregulated through the epithelial–mesenchymal transition (EMT) process [6]. EpCAM is not the only possible marker for CTC detection. PD-L1, for instance, is an immune checkpoint protein that is often overexpressed in various cancers, enabling tumor cells to evade immune detection by binding to the PD-1 receptors on T-cells. Targeting PD-L1 can improve the sensitivity of CTC detection and has been shown to enhance the capture rate when combined with other markers [7,8,9]. EGFR, or epidermal growth factor receptor, is a transmembrane protein involved in cell growth and differentiation. This protein is frequently mutated or overexpressed in cancers such as NSCLC. Targeting EGFR with specific antibodies can also improve the detection and characterization of CTCs [10]. The malaria protein VAR2CSA and a derived recombinant subfragment rVAR2 bind to oncofetal chondroitin sulfate (ofCS). ofCS is a secondary glycosaminoglycan modification and has been reported to be a modification present on the cell surface of tumor cells of all origins [11]. PDL-1, EFGR, and rVAR2 all represent an alternative to EpCAM in the enrichment of CTCs. Another option is to combine multiple markers to improve capture efficiency. In practice, tumors exhibit heterogeneity and diverse marker expression profiles. In light of this, employing a multi-marker strategy for CTC capture represents a promising approach to enhance capture efficiency.

To address variations in marker expression profiles due to different NSCLC subtypes and genetic backgrounds, and their potential impact on the effectiveness of the capture approach, we evaluated the expression of EpCAM, PD-L1, EGFR, and ofCS (rVAR2 binding target) across various cell lines. We selected four NSCLC cell lines—NCI-H1563, A549, NCI-H1792, and NCI-H661—that exhibit a broad range of EpCAM expression levels, from low (763) to high (91053), and investigated the complementary expression of other markers. Using this selection, we employed the previously reported magnetic nanoparticles (NC@silica-SA) and the FETCH separation system to evaluate the effect of combining anti-EpCAM and anti-ofCS (rVAR2) to enhance the recovery efficiency of NSCLC cell lines from blood [12].

## 2. Results

The experimental schematic is shown in Figure 1.

### 2.1. Comparison of ofCS, EpCAM, PD-L1, and EGFR Expression on NSCLC Cell Lines

Since our enrichment is based on targeting cell surface markers, we first measured the expression of different surface markers on the cell lines used. For this, we characterized the expression of ofCS using rVAR2 in addition to the frequently used EpCAM, PD-L1, and EGFR markers. To quantify the measured expression, we used a BD Quantibrite beads PE Fluorescence Quantitation kit. The expression of the markers on the four NSCLC cell lines used is shown in Figure 2. Clearly, high expression of ofCS is observed on the first three cell lines, while on the fourth cell line (NCI-H661), the EpCAM antigen is most actively expressed. EGFR expression was detected in medium to low amounts, while very low PD-L1 expression was detected in these four NSCLC cell lines. Given the high expression levels of ofCS and EpCAM, and the relatively low expression of PD-L1 and EGFR, the combination of rVAR2 and EpCAM was chosen for optimal targeting and enrichment of NSCLC cells.

### 2.2. Effect of Cell Fixation on ofCS Expression in NSCLC Cell Lines

Both live and fixed cells are used in clinical sample testing, depending on the scenario and accessibility; freshly collected blood is often used for same-day testing, whereas blood preserved in, for instance, CellSave preservative tubes allows researchers to process samples up to four days later. To evaluate the effect of preservation on the expression of ofCS, we measured ofCS expression on the surface of live cells and cells fixed in CellSave tubes. As depicted in Figure 3, both NCI-H1563 and A549 showed decreased ofCS expression on their surface after fixation with CellSave. Specifically, NCI-H1563 cells showed a 54% decrease, while A549 cells exhibited a 33% decrease in ofCS expression due to fixation. In our experiments, to preserve high ofCS expression, we used EDTA-collected blood spiked with live cells.

### 2.3. Comparison of Capture Efficiency in PBS, Whole Blood, and Lysed Blood in NSCLC Cell Line

The medium used (PBS, lysed blood, or whole blood) significantly affects the sensitivity of tumor cell enrichment. Our results on EpCAM-based cell capture show that, as expected, samples spiked in PBS yielded a higher capture efficiency than those in lysed blood (*p* < 0.05) and whole blood (*p* < 0.001) (Figure 4). The reason for this is that PBS does not contain any interfering substances that prevent binding or separation. As a result, A549 cells spiked into PBS achieved a capture efficiency of 83%. In contrast, when using lysed blood, which increases the presence of interfering factors such as proteins and leukocytes, the capture efficiency significantly decreased to 56% (*p* < 0.05). The capture efficiency further decreased to 22% when using whole blood (*p* < 0.001).

Although PBS results in the highest capture efficiency, it is not useful for evaluating methods aimed at clinical sample processing. As the recovery efficiency of A549 cells was higher in lysed blood, and this medium more closely resembles leukapheresis material, another potential way of further increasing the number of tumor cells detected [13], we used lysed blood as the medium for spiking cancer cells in further experiments.

### 2.4. Comparison of the Capture Efficiency Using Anti-EpCAM Antibody and rVAR2 in Lysed Blood

The expression of surface markers on tumor cells is closely related to their capture efficiency. Figure 5a,b show the relationship between marker expression and the resulting capture efficiency obtained with the four NSCLC cell lines when targeting EpCAM or ofCS, respectively. An increased capture efficiency correlates with higher expression levels for both EpCAM and ofCS. For instance, NCI-H1563 cells exhibit low expression of EpCAM, resulting in a recovery efficiency of only 6.8% when using the anti-EpCAM antibody. Conversely, NCI-H1563 cells have high ofCS expression, and their recovery efficiency using rVAR2 is, as a result, also high at 87.3%. Similarly, the A549 and NCI-H1792 cell lines are more effectively captured using rVAR2. On the other hand, the NCI-H661 cell line is captured very effectively using the anti-EpCAM antibody (94.6%) but shows very low capture efficiency using rVAR2 (26.6%). This indicates that the use of either anti-EpCAM or rVAR2 alone is insufficient to capture the full range of these four NSCLC cell lines.

### 2.5. Capture Efficiency When Using a Combination of Anti-EpCAM Antibody and rVAR2 Protein in Spiked Lysed Blood

To evaluate whether the combination of both markers can improve total capture efficiency, we compared the combined use of anti-EpCAM antibody and rVAR2 with the use of each marker individually. As shown in Figure 6a, the capture efficiencies based on the combined use of anti-EpCAM antibody and rVAR2 were 95.6%, 76.9%, 84.7%, and 95.7% for NCI-H1563, A549, NCI-H1792, and NCI-H661 cells, respectively. These capture efficiencies were higher than those achieved using either anti-EpCAM or rVAR2 alone.

An increase in general capture efficiency raises concerns about potential increases in nonspecific capture. As shown in Figure 6b, although there was a slight elevation in nonspecific capture with the combined use of anti-EpCAM antibody and rVAR2, it was not statistically different from the nonspecific capture observed with the use of anti-EpCAM antibody or rVAR2 alone. Therefore, the combined use of anti-EpCAM antibody and rVAR2 for the recovery of the four NSCLC cell lines is a feasible strategy.

Because both inter- and intra-patient heterogeneity in protein expression patterns are expected, we calculated the average capture efficiency of the four cell lines to assess capture of a heterogenous cell population. Figure 7 shows that the average capture efficiency using the combination of anti-EpCAM antibody and rVAR2 increased to 88.2% from 40.6% and 56.5% when using anti-EpCAM antibody or rVAR2 alone, respectively. Meanwhile, the non-specific binding increased to 3.9 from 2.6 and 3.4, and the enrichment factor increased to 23.2 from 15.6 and 16.5. This indicates a substantial improvement in capture performance when using the combination of anti-EpCAM antibody and rVAR2.

## 3. Discussion

The number of CTCs detected in metastatic non-small-cell lung cancer (NSCLC) patients is significantly lower as compared with metastatic prostate and breast cancer [14], while the survival prospect for NSCLC patients is dismal. This raises the question whether CTCs are indeed less frequent or simply missed by the detection method. The most frequently used method to identify CTCs is CellSearch, which is based on immunomagnetic enrichment of cells expressing EpCAM followed by immunofluorescence detection of cytokeratin and absence of the leukocyte antigen CD45. EpCAM is expressed on a large proportion of cells of epithelial origin, but not on the surface of blood cells, which makes it an attractive target for immunomagnetic enrichment of CTCs [15,16,17]. However, recent studies have revealed low or absent EpCAM expression in certain cancers [18,19]. To enhance EpCAM-based isolation, other markers such as PD-L1, EGFR, MUC1, and HER2 are often employed or combined [9,20]. In 2015, Salanti et al. introduced rVAR2 as a possible pan-cancer recognition protein, potentially serving as a universal targeting agent for over 95% of tumor cell lines [11]. Later, Agerbæk et al. demonstrated that rVAR2 binding to CTCs is independent of EpCAM expression and unaffected by EMT and MET [21]. Here, we evaluated the expression of PD-L1, EGFR, rVAR2, and EpCAM on different NSCLC cell lines and concluded that the combined use of rVAR2 and EpCAM enhances the efficiency of immunomagnetic enrichment for tumor cells with varying levels of target expression. To improve the capture of cells expressing low numbers of the target antigens, we used the FETCH system in combination with NC@silica-SA ferrofluids to separate CellTracker-labeled cells from samples incubated with biotinylated antibodies or rVAR2.

We separately evaluated the capture efficiency of ofCS and EpCAM for various NSCLC cell lines and concluded that neither EpCAM nor ofCS alone can efficiently capture all four NSCLC cell lines, highlighting the advantage of using both markers in combination (Figure 5). Combining anti-EpCAM and rVAR2 significantly improved the capture efficiency across all tested cell lines, as shown in Figure 6a. It has been shown that a large heterogeneity exists in marker expression both within and between patients. Although our cell lines cover a range of expressions, their resemblance to actual CTC is not known and will greatly differ from patient to patient. In these cell lines, the combined approach can potentially mitigate the limitations associated with the heterogeneity of single-marker expression, thereby resulting in an increased CTC capture in NSCLC patients. The limited and non-significant increase in nonspecific capture observed with the combined approach suggests that this strategy enhances capture efficiency without compromising specificity. The expression of EpCAM on normal lung epithelial cells is unlikely to significantly contribute to nonspecific capture in blood-based assays, as these cells are not typically present in circulation. Nonspecific capture, when it occurs, can be attributed to other non-tumor cells, such as CD11b+ cells, as noted in previous studies on EpCAM-based CTC detection [22]. Importantly, ofCS, the target of rVAR2, is absent in normal lung tissue, as shown by Salanti et al. [11], ensuring the specificity of rVAR2 for cancerous cells.

Applying this dual-marker approach to clinical blood samples presents specific challenges that need to be addressed. Evaluation of the assay in NSCLC patients is now needed to determine whether the increase in CTC yield observed in cell lines also leads to an increase in the number of CTCs detected in NSCLC patients. One caveat is that the standard detection of clinical CTCs is based on cytokeratin staining. Decreased or limited cytokeratin expression in NSCLC will lead to a lack of CTC detection, even if they are present in the rVAR2/EpCAM-enriched sample. Expanding the reagents used for CTC detection may mitigate this issue. Another limitation of the assay might be the effect of fixation of the blood on the binding of rVAR2, as this can lead to a reduction in capture efficiency. In this research, unfixed samples were used, which must be processed within 24 h of blood draw. This requirement poses a challenge for broader clinical implementation. Furthermore, variability in marker expression among patient-derived CTCs and the potential impact of blood preservation methods on capture efficiency require further optimization and validation in clinical settings.

## 4. Materials and Methods

### 4.1. Cell Cultures and Blood Sample Collection

Lung cancer cell lines (NCI-H1563, NCI-H460, A549, and NCI-H1792) and prostate cancer cell lines (PC3 and LNCaP) were cultured in an RPMI 1640 medium (Sigma-Aldrich, St. Louis, MO, USA) supplemented with 10% fetal bovine serum, L-glutamine, penicillin, and streptomycin. These cell lines were originally obtained in the context of the IMI CANCER-ID project. Cells were incubated at 37 °C in a humidified atmosphere and dissociated upon reaching approximately 80% confluence using Cellstripper (Corning Life Technologies, Tewksbury, AZ, USA). Cellstripper is a non-enzymatic cell dissociation solution that gently dislodges adherent cells in culture. Its advantage lies in its formulation with a proprietary mixture of chelators, which reduces the risk of cell damage associated with protein digestive enzymes. All cell lines used in this study were routinely monitored for mycoplasma contamination and tested negative.

Blood samples were collected from healthy volunteers in EDTA blood collection tubes at the University of Twente. In accordance with the Declaration of Helsinki, written informed consent was obtained from all volunteers, and the blood collection procedure was approved by the local Medical Research Ethics Committee (METC Twente).

### 4.2. Optimization of the Concentration of rVAR2 and Characterization of ofCS Expression in NSCLC Cell Lines

Recombinant VAR2CSA protein (rVAR2) with the structure SpyTag-DBL1-ID1-DBL2-ID2a-V5-6xHis [21] and recombinant SpyCatcher protein were produced and quality-controlled at the University of Copenhagen, Denmark as previously described [23]. For rVAR2, the DBL1-ID1-DBL2-ID2a sequence serves as the primary structure for recognizing and binding to oncofetal chondroitin sulfate (ofCS) [24]. The 6xHis tag is for purification purposes, and the V5 tag can be detected using an anti-V5 antibody labeled with phycoerythrin (PE). Additionally, the N-terminal SpyTag can be utilized to react with biotinylated SpyCatcher.

To assess ofCS expression on tumor cells, we employed a PE-labeled anti-V5 antibody (Invitrogen, Carlsbad, CA, USA). A total of 100,000 cancer cells (LNCaP and PC3 cells) were incubated with varying concentrations of rVAR2 (ranging from 25 nM to 200 nM) for 30 min at 4 °C in Dulbecco’s phosphate-buffered saline (DPBS) buffer containing 5% fetal bovine serum (FBS) and 1 mM ethylenediaminetetraacetic acid (EDTA). Following this incubation, the cells underwent two rounds of washing and were subsequently incubated with PE-labeled anti-V5 antibodies (1:20) for an additional 30 min at 4 °C. Finally, the cells were subjected to two more washing steps using a buffer containing 0.1% bovine serum albumin (BSA) and 2 mM EDTA in DPBS and were analyzed using flow cytometry (FACS Aria II, BD Biosciences, San Jose, CA, USA) to assess staining intensity.

To characterize ofCS expression in four NSCLC cell lines (NCI-H1563, A549, NCI-H1792, and NCI-H661), these cells were incubated with the optimized rVAR2 concentration. The experimental procedures remained consistent with the previously optimized protocol. Finally, the PE fluorescence signals obtained from all cells were quantified using a PE Fluorescent Quantitation Kit (BD Biosciences, San Jose, CA, USA).

### 4.3. Characterization of EpCAM, PD-L1, and EGFR Expression in NSCLC Cell Lines

To characterize EpCAM expression in four NSCLC cell lines (NCI-H1563, A549, NCI-H1792, and NCI-H661), 100,000 tumor cells from each cell line were incubated with a 1:5 dilution of anti-EpCAM-PE (SAB4700425-100TST, Sigma) for 30 min at 37 °C in DPBS/1% BSA buffer. Following this incubation, the cells were subjected to two rounds of washing and then analyzed using flow cytometry to assess staining intensity.

The EpCAM antibody is directly fluorescently labeled with PE, whereas the PD-L1 and epidermal growth factor receptor (EGFR) antibodies are not fluorescently labeled and therefore require the use of fluorescently labeled secondary antibodies. To characterize PD-L1 and EGFR expression in the four NSCLC cell lines, 100,000 tumor cells of each cell line were incubated with a 1:50 dilution of anti-PD-L1 (host: human cell, Miltenyi Biotec, Germany) or anti-EGFR (host: human cell, Novus Biologicals, Centennial, CO, USA) for 30 min at 37 °C in DPBS/1% BSA buffer. After two rounds of washing, cells were incubated for an additional 30 min at 37 °C in a 1:10 dilution of anti-human-PE antibody (Invitrogen, Carlsbad, CA, USA). After another two washing steps, the staining intensity was measured using flow cytometry, and the obtained fluorescence intensities were quantified using a BD Quantibrite beads PE Fluorescence Quantitation kit (BD Biosciences, San Jose, CA, USA). The mean fluorescence intensity of the isotype control group was subtracted from the measured signal to correct for background fluorescence.

### 4.4. Effect of Fixative on ofCS Expression in NSCLC Cell Lines

Blood samples are typically collected in either EDTA blood collection tubes or tubes containing a fixative, such as CellSave. Fresh blood specimens require immediate processing, preferably on the same day of collection, whereas blood collected in CellSave tubes can be processed within a 4-day window. To determine if preservation of samples is feasible when using rVAR2, we investigated whether different preservation methods impact the targeted ofCS expression. For this purpose, we characterized the expression of ofCS on the surface of fresh live cells and cells fixed in CellSave tubes for 24 h. The specific experimental steps are the same as in Section 2.3.

### 4.5. Comparison of Capture Experiments in PBS, Lysed Blood, and Whole Blood

The effectiveness of magnetic separations, among other factors, depends on the media used. In this study, we explored PBS, lysed blood, and whole blood as potential media for capture experiments.

The steps for lysing blood are as follows: for every 1 mL of blood, 15 mL of lysis buffer (10^−4^ M EDTA, 10^−3^ M KHCO_3_, 0.17 M NH_4_C1 aqueous solution, pH = 7.3) is added. After incubating at room temperature for 5 min, the mixture is centrifuged at 500 RCF for 5 min to remove the supernatant. Subsequently, the pellet is resuspended in 14 mL RPMI 1640 and centrifuged at 500 RCF for another 5 min. This process is repeated at least twice. Finally, the pellet is resuspended in 1 mL DPBS/1% BSA suspension.

A549 cells were initially stained with CellTracker Deep Red (Thermo Fisher Scientific, Waltham, MA, USA) at a concentration of 50 µM and Hoechst 33342 at a concentration of 5 µg/mL. Simultaneously, 3 mL of PBS, lysed blood, or whole blood was incubated with Hoechst 33342. Approximately 15,000 A549 cells were then added to each sample. Following this, 4.97 μL of 0.604 mg/mL biotinylated anti-EpCAM (clone VU1D9) antibody was added to the spiked sample resulting in a final labeling concentration of 1 μg/mL. The mixture was incubated at 37 °C for 45 min. Next, the samples were divided into three replicates, each consisting of a 1 mL aliquot. Each aliquot was incubated with 150 μg/mL of NC@silica-SA for 30 min. The mixtures were thoroughly mixed and subsequently incubated at room temperature for 30 min on a roller mixer. Finally, the tumor cells were recovered using the FETCH system. The specific FETCH separation steps and flow cytometry detection steps were previously described [12].

To assess the cell concentration in the starting samples, *reference* samples were prepared by spiking the same volume of tumor cells into an equivalent volume of media. The calculation of capture efficiency was carried out as follows:(1)$$Capture\;efficiency\;\left(\%\right)=\frac{Captured\;cells}{Reference}\times 100\%$$
where “*Captured cells*” refers to either the number of white blood cells or tumor cells captured, and “*Reference*” refers to either the number of white blood cells or tumor cells enumerated in the *references*, depending on whether the specific capture efficiency for tumor cells or the non-specific capture efficiency for white blood cells is calculated.
(2)$$Capture\;purity\left(\%\right)=\frac{Captured\;tumor\;cells}{All\;captured\;cells}\times 100\%$$
where “*All captured cells*” refers to the sum of specifically captured tumor cells and non-specifically captured white blood cells.
(3)$$Enrichment\;factor=\frac{Purity\;of\;enriched\;sample}{Purity\;of\;reference}$$
where “*Purity of reference*” refers to the number of spiked tumor cells divided by the sum of spiked tumor cells and white blood cells in the *reference* sample.

### 4.6. Comparison of the Capture Efficiency of Anti-EpCAM Antibody and SpyCatcher-rVAR2 or Their Combination in Spiked Lysed Blood

To compare the capture efficiency of different cell lines using anti-EpCAM and biotinylated-rVAR2, the four NSCLC cell lines (NCI-H1563, A549, NCI-H1792, and NCI-H661) were first pre-stained with CellTracker Deep Red and then added to separate 3 mL lysed blood samples from a single donor. The experimental procedure for labeling using biotinylated anti-EpCAM antibody was performed according to Section 2.5.

When using biotinylated rVAR2 to isolate tumor cells from blood, biotinylated rVAR2 was first prepared by reacting SpyTag-rVAR2 with biotinylated-SpyCatcher in PBS for one hour at room temperature. The details were described in previous work [23]. Following this, biotinylated rVAR2 (working concentration of 100 nM) was added to the spiked cell sample, and the mixture was incubated at 4 °C for 45 min. When combining both EpCAM and rVAR2, final concentration of 100 nM biotinylated SpyCatcher-rVAR2 and 1 μg/mL biotinylated anti-EpCAM (clone VU1D9) antibody were added to the 3 mL spiked samples, which were suspended in DPBS with 5% FBS and 1 mM EDTA, and the mixture was incubated at 4 °C for 45 min.

After the incubation, the samples were divided into 1 mL aliquots, and each aliquot was incubated for 30 min with 150 μg/mL of streptavidin (SA) coated magnetic nanoparticles (NC@silica-SA), with three replicates. The details of preparing NC@silica-SA are described in previous work [12]. The mixtures were thoroughly mixed and subsequently incubated at 4 °C for 30 min on a roller mixer. Subsequently, magnetic separation was performed using the FETCH system. The details of preparing the FETCH system are described in previous work [25]. The number of captured white blood cells and tumor cells was determined using flow cytometry.

## 5. Conclusions

In summary, we present a method for the enrichment of NSCLC cells from lysed blood samples using rVAR2 and anti-EpCAM antibody, leveraging our NC@silica-SA beads and the FETCH system. The combined use of rVAR2 and anti-EpCAM antibodies significantly enhanced tumor cell recovery compared with using either alone. The increased capture efficiency achieved with this combination of markers highlights the potential of a multi-marker approach in CTC detection, enabling the capture of a broader range of tumor cell phenotypes. Future research should validate these findings with clinical samples and explore integrating additional markers to further enhance CTC detection sensitivity.

## Figures and Tables

**Figure 1 ijms-25-09816-f001:**
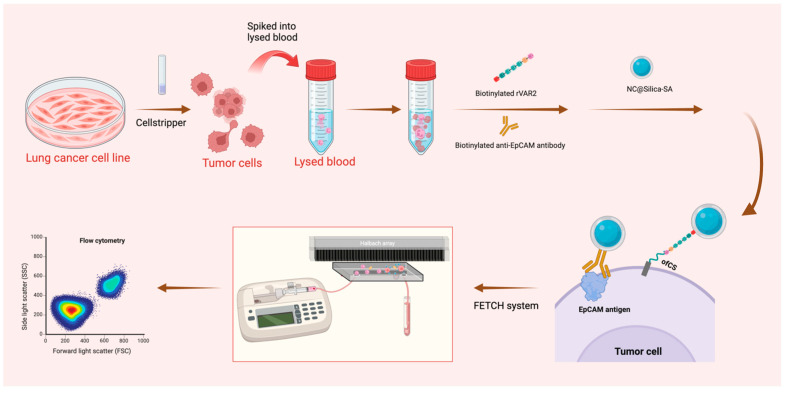
Schematic representation of the sample preparation and capture process. Lung cancer cell lines were pre-stained and detached using Cellstripper, then spiked into lysed blood. Biotinylated anti-EpCAM antibody and biotinylated rVAR2 were added, followed by washing steps post-incubation. NC@silica-SA was introduced to bind to the biotinylated tumor cells. The samples were then processed through the FETCH system, which captured NC@silica-SA bound cells, while other cells were discarded into the waste tube. The captured samples were subsequently enumerated using a flow cytometer.

**Figure 2 ijms-25-09816-f002:**
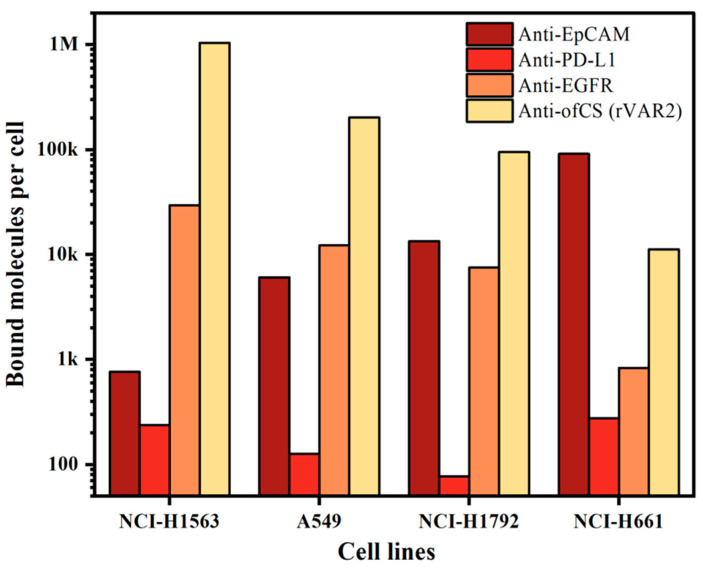
Quantification of bound molecules per cell for different molecules (anti-EpCAM, anti-PD-L1, anti-EGFR, and rVAR2) across four NSCLC cell lines (NCI-H1563, A549, NCI-H1792, and NCI-H661).

**Figure 3 ijms-25-09816-f003:**
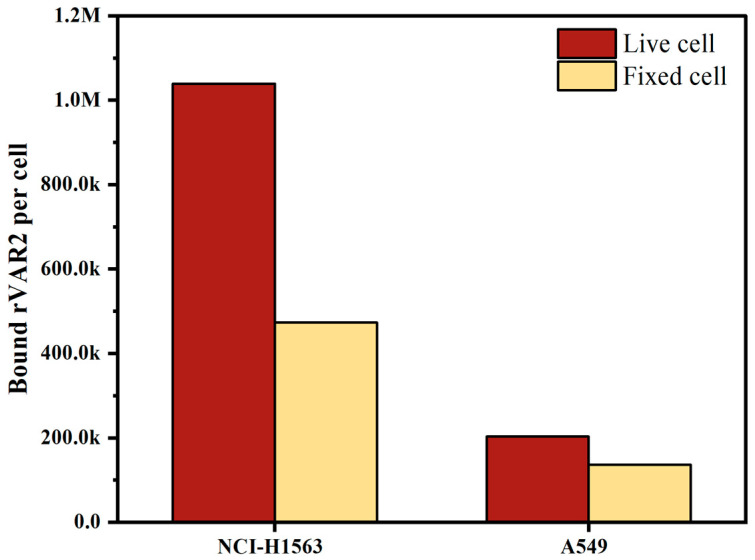
Comparison of oncofetal chondroitin sulfate (ofCS) expression on the surface of NSCLC cell lines (NCI-H1563 and A549) between live cells and cells fixed with CellSave for 24 h.

**Figure 4 ijms-25-09816-f004:**
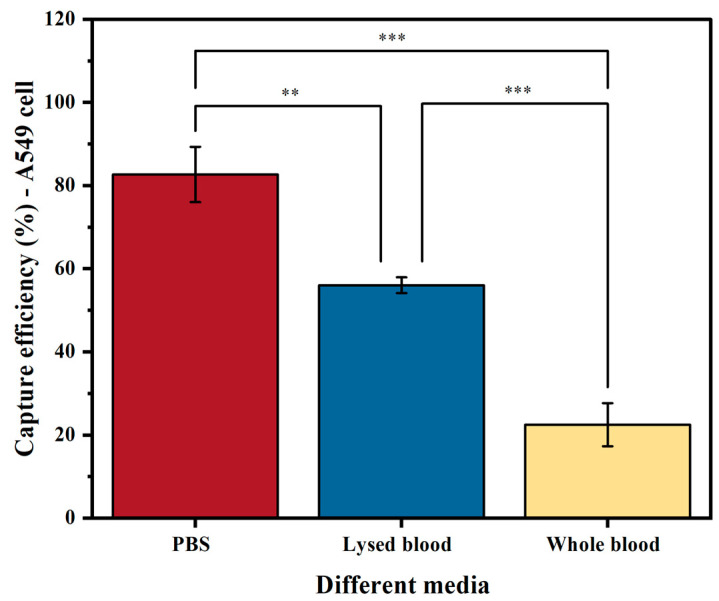
Capture efficiency of the A549 cell line using biotinylated anti-EpCAM antibody in three different media: PBS buffer, lysed blood, and whole blood. The columns represent the mean capture efficiency, and the whiskers indicate the standard deviation (SD) with *n* = 3. Statistical significance is denoted by ** (*p* < 0.05) and *** (*p* < 0.001).

**Figure 5 ijms-25-09816-f005:**
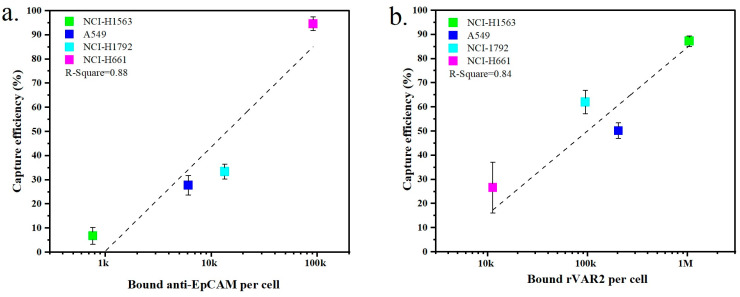
Comparison of the capture efficiency of NSCLC cell lines in lysed blood using (**a**) anti-EpCAM antibody or (**b**) rVAR2. Each data point represents the mean capture efficiency for a specific cell line, with whiskers indicating the standard deviation (SD) from three replicates (*n* = 3). The cell lines tested include NCI-H1563, A549, NCI-H1792, and NCI-H661. The correlation between capture efficiency and the number of bound molecules per cell is shown, with R-squared values indicating the fit of the regression lines.

**Figure 6 ijms-25-09816-f006:**
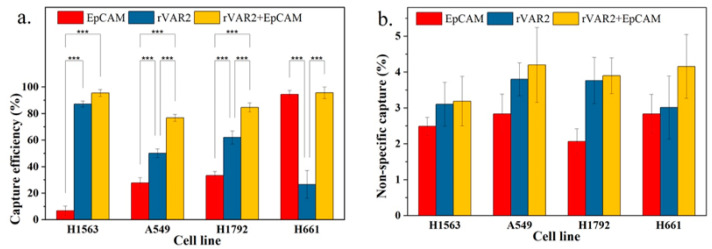
(**a**) Capture efficiency of NSCLC cell lines in spiked lysed blood using anti-EpCAM antibody, rVAR2, or their combination. (**b**) Nonspecific capture in spiked lysed blood for the same conditions. Columns represent the mean capture efficiency and nonspecific capture, with whiskers indicating the standard deviation (SD) for *n* = 3. Statistical significance is denoted by *** (*p* < 0.001), indicating that the combination of anti-EpCAM and rVAR2 significantly improves capture efficiency compared with using either marker alone.

**Figure 7 ijms-25-09816-f007:**
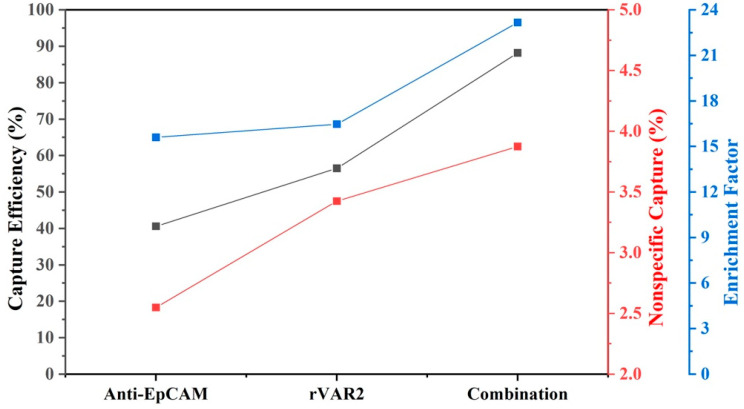
Average capture efficiency, nonspecific capture, and enrichment factor for four NSCLC cell lines using different enrichment conditions: anti-EpCAM, rVAR2, and their combination. The results assume equal amounts of each cell line prior to enrichment. The left *y*-axis (black) represents capture efficiency (%), the middle *y*-axis (red) represents nonspecific capture (%), and the right *y*-axis (blue) represents the enrichment factor. The combination of anti-EpCAM and rVAR2 shows improved performance across all metrics compared with using either marker alone.

## Data Availability

The data supporting in manuscript are available in the article and Appendix A. All other data are available from the corresponding author upon reasonable request.

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
