# Peer review of "Combining rVAR2 and Anti-EpCAM to Increase the Capture Efficiency of Non-Small-Cell Lung Cancer Cell Lines in the Flow Enrichment Target Capture Halbach (FETCH) Magnetic Separation System"

_ijms, 2024, doi:10.3390/ijms25189816_

Round 1

Reviewer 1 Report

Comments and Suggestions for Authors

In this study, the author introduced a promising method to enhance the efficiency of capturing circulating tumor cells (CTCs) from non-small cell lung cancer (NSCLC) patients.  The research demonstrates that the combination of anti-EpCAM antibody and rVAR2 protein significantly improves the retrieval of NSCLC cell lines from lysed blood samples using the Halbach (FETCH) magnetic separation system.  This dual-marker strategy overcomes the limitations of relying exclusively on EpCAM for CTC enrichment.  The following issues need to be resolved before the article is published.

1. It is advisable to centrally align the image within the article for a more aesthetically pleasing and reader-friendly layout.

2. The study is limited to only four NSCLC cell lines, which may not fully represent the diversity observed in patient tumors.  Variations in marker expression profiles due to different NSCLC subtypes and genetic backgrounds could potentially impact the effectiveness of the dual-marker approach.  Whether the authors selected these four types of cells, and what considerations were taken into account?

3. While the study demonstrated an increase in nonspecific capture with the dual-marker approach, the potential impact on the expression of ofCS and EpCAM in normal lung cells remains incompletely explored.

Comments on the Quality of English Language

Minor editing of English language required.

Author Response

Comment 1: It is advisable to centrally align the image within the article for a more aesthetically pleasing and reader-friendly layout.

Response 1: Thank you for this suggestion. We have centrally aligned the images in the manuscript to enhance the visual presentation and improve readability.

Comment 2: The study is limited to only four NSCLC cell lines, which may not fully represent the diversity observed in patient tumors.  Variations in marker expression profiles due to different NSCLC subtypes and genetic backgrounds could potentially impact the effectiveness of the dual-marker approach.  Whether the authors selected these four types of cells, and what considerations were taken into account?

Response 2: Thank you for your insightful comment. We agree that variations in marker expression profiles due to different NSCLC subtypes and genetic backgrounds could potentially impact the effectiveness of the dual-marker approach. We have further clarified the considerations behind the selection of cell lines in the introduction section, as noted below.

Page 2, line 75

“To address variations in marker expression profiles due to different NSCLC subtypes and genetic backgrounds, and their potential impact on the effectiveness of the capture approach, we evaluated the expression of EpCAM, PD-L1, EGFR, and ofCS (rVAR2 binding target) across various cell lines. We selected four NSCLC cell lines—NCI-H1563, A549, NCI-H1792, and NCI-H661—that exhibit a broad range of EpCAM expression levels, from low (763) to high (91053), and investigated the complementary expression of other markers. Using this selection, we employed the previously reported magnetic nanoparticles (NC@silica-SA) and the FETCH separation system to evaluate the effect of combining anti-EpCAM and anti-ofCS (rVAR2) to enhance the recovery efficiency of NSCLC cell lines from blood.”

Comment 3: While the study demonstrated an increase in nonspecific capture with the dual-marker approach, the potential impact on the expression of ofCS and EpCAM in normal lung cells remains incompletely explored.

Response 3: Thank you for this important observation. The expression of EpCAM on normal lung epithelial cells has minimal impact on nonspecific capture in our dual-marker approach because normal lung cells are not typically found in blood circulation. While nonspecific capture may occur due to cross-reactivity with other non-tumor cell types, such as CD11b+ cells, EpCAM-based CTC capture methods have reported these occurrences and highlight the need to distinguish true CTCs from such events.

In contrast, ofCS, which is the binding target of rVAR2, is absent in normal lung tissue, as demonstrated by Salanti et al. (2015) and Oo et al. (2021). This ensures that rVAR2 targets cancer cells specifically, thereby minimizing nonspecific capture and increasing the specificity of our dual-marker approach.

We further clarified in the discussion section, as below.

Page 9, line 276

“The expression of EpCAM on normal lung epithelial cells is unlikely to contribute significantly to nonspecific capture in blood-based assays, as these cells are not typically present in circulation. Nonspecific capture, when it occurs, can be attributed to other non-tumor cells, such as CD11b+ cells, as noted in previous studies on EpCAM-based CTC detection[27]. Importantly, ofCS, the target of rVAR2, is absent in normal lung tissue, as shown by Salanti et al. and Oo et al.[14,28], ensuring the specificity of rVAR2 for cancerous cells.”

Reviewer 2 Report

Comments and Suggestions for Authors

Dear authors of “Combining rVAR2 and Anti-EpCAM to Increase the Capture Efficiency of Non-Small Cell Lung Cancer Cell Lines in the Flow Enrichment Target Capture Halbach (FETCH) Magnetic Separation System”,

Thanks for your contribution to this field. This is an interesting article aimed at identifying new biomarkers in the capture of circulating tumor cells in the diagnostic of metastatic non-small cell lung cancer (NSCLC).

The manuscript is well written. Research is well organized, with a robust analysis method, and the obtained results are convincing. The findings are of interest for lung cancer research. They are well-discussed regarding the actual literature, and some limitations are included.

Nevertheless, I notice some minor points that need to be adjusted.

1-      Optimization of rVAR2 concentration has been done with prostate cancer cells, which is quite questionable since the aim is to capture lung cancer cells. This optimization, even necessary, doesn’t provide robust data in this manuscript, and I may suggest removing it to avoid the confusion.

2-      The final use of this capture system will be for human blood. It is a shame that this part is not considered or mentioned (other limitations) in the discussion.

 Looking forward to seeing your modifications,

All the best,

Author Response

Comment 1: Optimization of rVAR2 concentration has been done with prostate cancer cells, which is quite questionable since the aim is to capture lung cancer cells. This optimization, even necessary, doesn’t provide robust data in this manuscript, and I may suggest removing it to avoid the confusion.

Response 1: Thank you for your suggestion. We agree with your assessment and have moved the section on rVAR2 concentration optimization using prostate cancer cells to the supplementary material.

Comment 2: The final use of this capture system will be for human blood. It is a shame that this part is not considered or mentioned (other limitations) in the discussion.

Response 2: Thank you for bringing this to our attention. We have now included below into the discussion on the application of our capture system in human blood, including potential challenges and limitations.

Page 9, line 283

“Applying this dual-marker approach to clinical blood samples presents specific challenges that need to be addressed… Furthermore, variability in marker expression among patient-derived CTCs and the potential impact of blood preservation methods on capture efficiency require further optimization and validation in clinical settings.”

Round 2

Reviewer 1 Report

Comments and Suggestions for Authors

The author has revised the manuscript in accordance with the reviewer's comments and may accept the manuscript in its present form.

Comments on the Quality of English Language

None